# Stigma, coping strategies, and their impact on treatment and health outcomes among young men living with HIV in Vietnam: A qualitative study

**Nhu Kieu Tran**[1]*, **Bach Ngoc Vu**[1], **Jordan Susa**[2], **Mary DeSilva**[2]

**1** Institute for Social Development Studies, Hanoi, Vietnam, **2** Center for Excellence in Public Health, University of New England, Portland, Maine, United States of America

* trankieunhu@isds.org.vn

## Abstract

### Background

Stigma affects persons living with HIV in myriad ways, including mental health, adherence to antiretroviral therapy, and retention in care, and may manifest at inter- and intra-personal levels. Youth are particularly vulnerable; those in vulnerable groups may experience multiple stigmas. In Vietnam, new HIV infections are rising among men in young age groups. To better understand the facets of stigma experienced by young men living with HIV in Vietnam, we conducted a qualitative study with youth and clinicians.

### Methods

We conducted in-depth interviews with ten youth and two clinicians in Hanoi, querying experiences of inter- and intra-personal stigma, coping strategies, and disclosure. As a framework for further research, we developed a conceptual model based on our findings and the published evidence base which portrays interactions among HIV-related stigma, coping strategies, and ART adherence, care engagement, and health outcomes.

### Results

Common themes that emerged from interviews with youth included extensive internalized/self-stigma and perceived stigma, yet limited experienced interpersonal stigma due to non-disclosure and avoidant coping strategies. Within different types of relationships or contexts, youth used different strategies. Non-disclosure with family, friends, and workplaces/school, and avoidance of romantic relationships and health care were common. Mental health and social support appeared to be mediating factors between coping strategies and health outcomes.

### Conclusions

Validation of this model of mechanisms of the impact of stigma for youth will require further research with larger samples. In the meantime, public campaigns to increase public

**Data Availability Statement:** Data are available from the Institutional Data Access Center of the Institute for Social Development Studies (ISDS) for researchers who meet the criteria for access to

confidential data. Contact via the ISDS Ethical Committee at the email: trankien@isdsorg.vn; Address: Lo 81-TT4, My Dinh Song Da Urban Areas, My Dinh 1, Nam Tu Liem, Ha Noi, Viet Nam; Phone:+84.243.7820058.

**Funding:** NKT and MBD received the grant number R21TW011085 from the United States National Institutes of Health, Fogarty International Center and Eunice Kennedy Shriver National Institute of Child Health and Human Development. The website of the funder is: https://www.fic.nih.gov/Grants The funder had no role in the study design, data collection and analysis, decision to publish, or preparation of the manuscript.

**Competing interests:** The authors have declared that no competing interests exist.

awareness related to HIV should be implemented in Vietnam. Critical support for youth and their mental health should involve approaches tailored to the individual, taking into account context and personal capacity, including adequate time to prepare psychologically for disclosure. Some strategies for safe and effective disclosure are suggested.

## Introduction

HIV-related stigma, a critical challenge for persons living with HIV (PLHIV), has adverse effects on mental health, health care access, and HIV care including adherence to antiretroviral therapy (ART) and retention in care [1–3]. Stigma may act at both inter- and intra-personal levels to discredit and devalue an individual's identity [4]. At the interpersonal level, *enacted stigma* refers to overt acts of hostility and discrimination toward stigmatized persons. At the intrapersonal level, stigma may manifest as *perceived stigma*, an individual's subjective perception of their stigmatized status, or *internalized (self) stigma*, the incorporation of the devalued attribution of society into one's sense of self. Studies have documented that PLHIV in resource-limited settings like Vietnam, Indonesia, and Thailand experience stigma and discrimination from families, communities and health care settings [5–9].

Adolescence and young adulthood are characterized by risk-taking behaviors, strong needs for peer relationships and acceptance, and greater vulnerability to mental health difficulties [10]. These developmental characteristics may worsen the impact of stigma on youth living with HIV (YLHIV). Adverse mental health outcomes are prevalent among YLHIV [11–14], and have generally been found to be substantially higher than among counterparts not living with HIV across contexts in the few studies where comparison groups are included [14]. For example, a study in the United States found that 21% of YLHIV reported mood disorders, significantly higher than the 5% of youth not living with HIV [15]. In Zambia, adolescents living with HIV had a significantly higher prevalence of mental health problems (29.1% compared to 16.5% in a community comparison group), particularly emotional symptoms (31.5% vs. 11.2%) and peer problems (41.8% vs. 9.2%) [16].

The dynamics and effects of stigma among youth living with HIV (YLHIV) are less well documented than among adults. A few studies in resource-limited settings like Uganda and Thailand have found internalized stigma, perceived stigma, and psychological difficulties to be common among YLHIV [5, 17]. Young people in vulnerable groups may also experience multiple stigmas, with the stigma brought by identifying with a vulnerable group compounded by the stigma of HIV. In a cohort of adolescents in South Africa, for example, multiple forms of discrimination and the resultant internalized stigma were found to contribute to low retention in HIV care [18].

To cope with perceived impacts of stigma, PLHIV may employ various strategies. Choosing whether to disclose one's HIV status is often a central aspect. The Disclosure Processes Model (DPM) framework describes the mechanisms by which disclosure or non-disclosure is chosen as well as the mediating processes of alleviating inhibition, social support, and changes in social information [19]. According to this framework, individuals with an avoidant approach would choose not to disclose their HIV status to others.

Non-disclosure, a common approach [20, 21], limits YLHIV's opportunities for social support which may negatively affect ART adherence and engagement in care [21, 22]. Non-disclosure within sexual relationships may threaten condom negotiation and partners' HIV testing [22, 23]. Moreover, concealment itself may cause distress for youth, as they have to hide and/

or tell lies regarding their medication-taking and keep all negative emotions regarding their illness to themselves [24], which research has shown to be stressful and may lead to psychological problems and physical illness [25].

Despite the disadvantages of non-disclosure, the scientific evidence for the impact of disclosure is mixed. Reviews about self-disclosure among YLHIV have shown that disclosure improved social support for ART adherence, access to care, financial support, psychological well-being, and emotional relief, and built trust in romantic relationships [26, 27]. These findings are in line with the DPM which suggests that alleviation of emotional inhibition mediates the effect of disclosure, and allows individuals to express real emotions and thoughts which accordingly bring about positive effects of disclosure [19]. However the reviews also found that some studies with YLHIV showed that disclosure was associated with reported internalized stigma, enacted stigma, feelings of isolation, and anxiety and depressive symptoms [26, 27], but not with increased condom use [26].

Avoidant coping strategies toward sexual relationships and health care are also common among PLHIVs [24]. They avoid sexual relationships due to fear of HIV transmission to a partner and fear of rejection and discrimination. As with non-disclosure, avoidance of relationships also tends to reduce social connections, and may increase the psychosocial risks of social isolation [28]. Some studies found that these psychological problems increased the risk of inconsistent condom usage for those engaged in sexual behavior [28]. The avoidance of accessing health care, caused by the fear of inadvertent disclosure through the health care system and the fear of shame when HIV status is disclosed, may compromise health outcomes for PLHIV. Avoidant coping strategies may be helpful in the short term, although the perceived threat persists and may affect PLHIV's functioning.

In Vietnam, approximately 215,000 people were living with HIV in 2019 [29], and although new HIV infections and AIDS-related deaths have plateaued in recent years in the country, the number of PLHIV is estimated to be 44 times greater than reported [30]. 40.1% of new HIV infections in Vietnam occur among individuals 16–29 years old [29], and MSM represent a large percentage of new infections. In 2015, HIV sentinel sero-surveillance estimated the HIV prevalence among MSM in Vietnam to be 5.2%; by 2017, prevalence had doubled (12.2%) [31]. HIV prevalence among MSM in Hanoi specifically increased from 9.4% in 2006 to 20% in 2010 [32]. Research has shown that unprotected sex, multiple sex partners, stigma, homophobia, substance use, and mental health difficulties are risk factors for HIV transmission among MSM in Vietnam [33–36]. MSM and MSM youth are considered a priority population for HIV prevention in the country's national HIV strategy [29, 37, 38].

Stigma research in Vietnam has documented a pervasive yet complex patchwork of HIV-related stigma, including within the healthcare system [6, 39]; stigma towards PWID [40, 41], and those involved in transactional sex work [42]. The experience of men living with HIV in Vietnam is affected by both societal perceptions of HIV and attitudes toward MSM [34, 43]. In the 2000's, homosexuality was considered to be a "social evil" *(tệ nạn xã hội)* by the government [44]. Social evils "run counter to the morality and fine customs and habits of the nation, adversely affect the health of the race, and the material and cultural life of The People" [45]. However, due to rapid socioeconomic transformation, knowledge of and attitudes towards homosexuality have improved, and expressions of sexual identity are more liberated. Nonetheless, stigma towards homosexuality remains common. Drug use, prostitution, and gambling are also considered social evils in Vietnamese culture. PLHIV are often assumed to be drug users or involved with sex work and/or homosexuality [37]. For MSM specifically, a study of healthcare workers in Hanoi and Ho Chi Minh City found extensive misperceptions of and stigma towards MSM [39]. A study of HIV testing and treatment access among adult MSM living with HIV in Vietnam found that intersecting stigma, discrimination, lack of social support,

non-disclosure of HIV status and sexual orientation, and mental health difficulties were barriers to accessing testing and treatment [46]. Little research in Vietnam in particular and in Asia in general has focused on stigma experienced by YLHIV or young MSM, or on coping strategies. In the recent global review of disclosure and mental health among YLHIV in which the majority of studies had been conducted in wealthy countries, only two studies were from Asian populations [26]. To our knowledge, no studies about stigma specifically among the young population living with HIV in Vietnam have been published. Thus, to better understand the facets of stigma experienced by young men who are members of vulnerable groups and living with HIV in Vietnam, we conducted a small qualitative study with YLHIV and clinicians in Hanoi. The objective was to describe youths' experiences of stigma, their coping strategies for dealing with stigma, and their impact on health outcomes of YLHIV. This study is part of a larger project that has developed a cognitive behavioral therapy- (CBT)-based intervention for youth ("Sống vui, Live happily") delivered by telephone that targets multiple facets of stigma. The preliminary efficacy of this intervention is currently being evaluated; findings will be presented upon completion.

## Materials and methods

### Participants and study design

This study was conducted in February 2020 in Hanoi, Vietnam, where 28,668 PLHIV resided in 2019 [30]. The study site was the Nam Tu Liem Outpatient Clinic (OPC) in Hanoi. The clinic serves approximately 1500 PLHIV, including 500 MSM, a few transgender individuals, 300 women, and 400 PWID. Nearly all of the patients in the study age range were men who have sex with men (MSM). The clinic has 10 staff: four nurses, three physicians, one pharmacist, and two medical technologists. Patients in ART care at the clinic generally attend the clinic once per month for general checkups and to obtain ART medication refills. HIV treatment is paid for from health insurance, out-of-pocket, or from donor-funded HIV/AIDS projects. Eighty percent of patients at the clinic have health insurance, and twenty percent pay out-of-pocket. Standard of care for PLHIV in Vietnam indicates CD4 testing at diagnosis, viral load (VL) testing after 6 and 12 months in treatment, then every 12 months. The OPC refers out for CD4 and VL testing.

We conducted in-depth interviews (IDIs) with youth and with clinic staff who provide ART care for youth at the clinic. Eligible youth were 18–24 years old and in ART care coordinated by the clinic, agreed to follow study procedures, and provided written informed consent. The two clinic staff members who work most closely with youth patients were invited to participate and provided written informed consent.

The qualitative study was conducted as part of the development process for a stigma reduction intervention, and the number of interviews was determined in large part by time and grant resources available. Given that common themes recurred frequently throughout interviews, we are confident that we came reasonably close to data saturation, whereby no further major themes were likely to be discovered [47].

The study was approved by institutional review boards at the Institute for Social Development Studies in Hanoi and the University of New England in the U.S.

### Data collection

Two female interviewers with background in social science research methods and experience working with PLHIV conducted IDIs in Vietnamese in private rooms at the clinic. The first author was one of these interviewers. Interviewers used a semi-structured interview guide which explored experiences living with HIV including disclosure, life activities, and sources of

social support; internalized stigma and perceived stigma; and experiences with treatment adherence. The interviews also probed for suggestions about adaptation of the CBT-based intervention concept to a phone-based modality acceptable to YLHIV to inform the broader project. The IDIs with clinic staff focused on experiences treating youth, and their perception of challenges faced by YLHIV including barriers to adherence, mental health challenges including disclosure, and suggested strategies for YLHIV experiencing mental health issues. Interviews lasted from 40 to 60 minutes and were audio-recorded and supplemented with written notes.

### Analytic methods

Audio-recordings were transcribed in Vietnamese by two researchers, one of whom was an interviewer. The transcripts and data interpretation were reviewed by both interviewers. When discrepancies were identified, the researchers consulted the Vietnamese transcripts together until consensus was reached regarding interpretation. The transcripts were then translated into English by a bilingual translator. English transcripts were reviewed by a second investigator for accuracy, then coded by one of the authors (JS) and analyzed using NVivo software and an iterative codebook [48].We used a thematic content analysis approach to identify important themes present in the transcripts [49]. We built an iterative codebook starting with the broad domains of inquiry as outlined in the interview guide, then as transcripts were coded in NVivo, continued to identify and create additional codes as they emerged in individual transcripts. After initial coding of all transcripts, the coder re-visited each transcript and applied all relevant codes from the comprehensive codebook that was developed in the process. Preliminary analysis revealed major themes, with further generation of sub-themes. We categorized responses by their frequency and also considered contrasting observations. The approach was primarily inductive and arose from the data, although some deductive elements were present, including initial broad domains as queried in the interview guide.

## Findings

### Overview

We conducted twelve IDIs, ten with youth and two with clinic staff. Mean age of youth participants was 22.4 years (SD 1.0 years); all were male (Table 1). Eight youth were MSM, one was a PWID, and one did not report identifying with any vulnerable group besides living with HIV. Five youth were students, and five were employed. The two staff interviewed included one physician and one nurse counselor. Table 2 presents coping strategies and disclosure status of the youth by relationships and setting.

Common themes that emerged from the interviews with youth included extensive internalized/self-stigma and perceived stigma, yet limited experienced interpersonal stigma due to non-disclosure and avoidant coping strategies. Within different types of relationships or contexts, YLHIV used different strategies. Below we present these findings in detail, organized primarily by type of relationship or context. Health staff provided additional context to experiences described by youth. Direct statements are provided below to illustrate findings.

### General experiences of stigma

**Psychological impact of learning HIV status.**   Participants' reactions to first learning their HIV status ranged from being shocked, disappointed, and worried, to being calm and prepared. Of the ten youth, three were shocked. Five others were not shocked, but felt disappointed, unconfident, and worried.

**Table 1. Characteristics of interview participants.**

| Characteristic, youth (N = 10) | N(%) or Mean (SD) |
|---|---|
| Gender (male) | 10 (100.0) |
| Age (years) | 22.4 (1.0) |
| Living situation | |
| alone | 1 (10.0) |
| with family | 5 (50.0) |
| with friend(s) | 4 (40.0) |
| Only child in family* | 4 (40.0) |
| Employment status | 5 (50.0) |
| Employed | 5 (50.0) |
| Student | 5 (50.0) |
| Group identification | |
| MSM | 8 (80.0) |
| PWID | 1 (10.0) |
| Other | 1 (10.0) |
| **Characteristic, clinic staff (N = 2)** | |
| Gender (female) | 2 (100.0) |
| Age (years) | 36.5 (0.7) |
| Position in clinic | |
| Clinician | 1 (50.0) |
| Counselor | 1 (50.0) |

* 4 participants volunteered this information; it was not asked explicitly.

The counselor noted that over her seven years' working with PLHIV, she had observed improvement over time in youths' knowledge and attitude when they first learn about their HIV status:

> The way people think about the disease and knowledge about the disease have improved. Cases where one feels so desperate to the point of suicide are not common anymore. However, [youth] still seem to lack deeper knowledge such as the low levels of transmission among PLHIV who are in ARV treatment and have undetectable viral load. (Counselor)

**Table 2. Coping strategies of youth participants (N = 10).**

| Coping strategies | N (%) |
|---|---|
| Non-disclosure to family | 7 (70.0) |
| Non-disclosure to friends | 6 (60.0) |
| Avoidance of romantic relationships (N = 9)* | 5 (55.6) |
| Non-disclosure to peers | 5 (50.0) |
| Non-disclosure in workplace or school | 10 (100.0) |
| Selective disclosure in health care context only if involving blood | 3 (30.0) |
| Avoidance of accessing healthcare (apart from HIV care) | 7 (70.0) |

* Sample includes one individual who characterized himself as asexual.

**Internalized stigma.**   Over time, the majority of youth participants (8/10) did report having incorporated the devalued attribution of society about their serostatus into their sense of self, however to a lesser extent than providers had observed in the past. Youth nonetheless still reported blaming themselves, losing self-confidence, and feeling guilty and worried about the future. One youth who contracted HIV by accident and felt that it was not his fault described his outlook:

> I contracted HIV by accident. It's not my fault. . .I just regretted a little, but not to the point where I have to change my mind about myself. (Asexual, 23 years old)

**Perceived stigma.**   Perceived stigma was common among the participants. Six participants directly attributed their lack of disclosure to fear of stigma and discrimination, unprompted. As one young man described,

> I have not disclosed to families, my friends. Only several people in the same situation know that I live with HIV, so I don't have experiences of stigma and discrimination. The fear of stigma and discrimination is the reason that I haven't disclosed. (MSM, 22 years old)

Participants also provided further context about the dynamics of stigma and discrimination. Several attributed discrimination to a lack of information or education.

> Because they [people] don't have enough information [about PLHIV], . . . they can't accept [us]. (MSM, 21 years old)

> . . . people now have a better view of LGBT, but people still have a bit of prejudice against people with HIV. Like me, if I did not have the knowledge, I would still have a prejudiced view against HIV, . . . If people do not have relatives with HIV, they will not go to learn about HIV at all. . . . (MSM, 23 years old)

However, one participant contended that discrimination remains pervasive, describing tacit behaviors by some whose social class might indicate a better-informed understanding of HIV and its stigma:

> It also happens in the educated class; they will not say anything but stay away from you. . . For teachers like me, if I disclose my status, the students' parents will not let their children learn with me. Students will look at me with a different view. (MSM, 22 years old)

**Dynamics underlying internalized stigma and perceived stigma.**   Besides fearing rejection and discrimination, youth also expressed worries about future hopes and plans regarding health, career, marriage, and having children.

> There are times that I feel unconfident, shame, disappointed. But it is not about me having lower value than my friends, because my friends do not know that I am living with HIV. I am just worried that over time, my health status is not as good as my friends' health. They can get married, [have] success in their life, have a good job (MSM, 21 years old)

The clinic physician elaborated:

Patients worry about their future. For example, some of our patients are MSM but still want to get married to a woman and have kids to maintain the family lineage. But they are afraid if they share with their female partners about their HIV status, the partners probably won't marry them. (Doctor)

Several participants described their HIV diagnosis as not only a failure for themselves, but also a failure for the family, even extended family:

They could send the information about me to the village. It would be not only my family knowing about [my status] but also the neighbors. . . I'm not worried for myself but I'm afraid they will talk about my parents. . .. I'm very scared that they will say that my family is not "*[phúc]*-blessed" [i.e., when a person does good things in his life, his descendants will have good luck or *[phúc]*. If a person has bad luck, his predecessor may have done bad things,]. . . I may not live in my homeland, but my parents are living there, so I'm afraid my parents will be affected by that. . . (MSM, 22 years old)

Moreover, several participants also expressed fear of causing family distress:

Because firstly I love my parents, secondly, they have high expectations for me. In my extended family, I was the first to study in university. So people expect a lot from me. I don't want to let people know my status. In general, it makes my parents sad and frustrated about me. The higher the expectation, the more disappointment if they learn [of my status]. (MSM, 22 years old)

Because if their parents know, the parents will be disappointed because parents always expect highly of [their children]. [The youth] are also afraid that their parents do not understand and still think that HIV is a social evil. (Doctor).

Being a son, especially being an only child, carries the additional pressure of feeling the responsibility of providing a descendent for the family. As one young man shared:

I am the only child, my parents urge me to get married, and have a child. I am afraid that there is no method completely safe to not transmit HIV to my child in the future. (MSM, 22 years old)

Thus, cultural gender norms may well add more psychological burden for YLHIV who are the only male child in a family.

The physician shared an observation related to MSM who may want multiple sexual partners:

Some patients shared with us that they had so many partners, that they couldn't remember them all. Regarding the characteristics of MSMs, unlike women who usually talk with partners to get to know them better before going out, MSM often go to various websites to search for sexual partners. If the sex is good enough, they will make another date. If not, after that sexual encounter, they will not contact the person again. They also usually contact each other by their fake name or nickname only. . .If they say they have HIV, they are afraid that their partners will not love them anymore, which means no more sexual partners. (Doctor)

## Coping strategies with families and their impact

Among the participants in the study, non-disclosure was the most frequently cited coping strategy. Seven of the ten YLHIV had not disclosed their status. Only one person had intentionally disclosed his status to his family, though two others had had their status inadvertently disclosed including the PWID whose family had learned from the rehabilitation center and a MSM whose mother had found out when cleaning his room.

As one young man explained,

Actually, they haven't disclosed yet. Like myself, my friends or family don't know about my status either. I could come out as a gay, but for my health status, I do not dare to disclose . . .I am not brave enough to sit down in front of my parents to declare that I have HIV. (MSM, 23 years old).

Another described the moment when he learned his status: "I feel sorry for my family, because at that moment I felt guilty. What will happen if my parents learn about it later?" (MSM, 22 years old).

Health staff also noticed the same situation. As the counselor explained,

Usually I think they are distressed, most of them haven't informed their family of their condition, especially parents. Most young people haven't had their own family so they still live with parents. Some do not, but they still usually don't inform their closest family members. (Counselor)

**Impact of non-disclosure.** Despite the perceived benefit of not disclosing to family in order to avoid negative reactions, youth experienced negative mental health effects of non-disclosure to family. As one participant related:

My friends hide their HIV status from their families. When their mood is down, or they feel irritated, sorry for themselves, and tired from side effects of medication, their families don't understand. They feel lonely. They have to find ways to hide their HIV status from their families, for example that they are taking medication. Hiding is always tiring. (MSM, 21 years old)

The counselor explained further:

The majority of people I've seen who haven't shared their situation with parents or lovers, their wives and children, deep down in their hearts, they always feel guilty, like they're dishonest with the person they really love. . . They are afraid that if their secret is discovered one day, their relationship and their career will disappear. That's why they always feel uncomfortable. (Counselor)

The health staff also explained how YLHIV could lose a source of treatment support if their family does not know about their HIV status.

MSM patients, in addition to HIV, often have many contagious diseases, such as syphilis and gonorrhea. They still have to pay for the treatment of other diseases. Many patients are students, and their financial resources are limited. Students live far away from their families,

so their diet is worse than if their families already know. When the family knows, the family will care for them better. (Doctor)

Thus, non-disclosure may affect diet and treatment for HIV as well as other comorbidities. The fear of being hurt emotionally or hurting parents emotionally by disclosing their HIV status made youth feel not brave enough to disclose. However, in the counselor's opinion, parents will accept their children regardless of their HIV status:

When they overcome that obstacle of sharing, their gain is huge. It doesn't count as a partner or lover who can leave them but their parents [will] never give them up. . . No matter how their children turn out, [parents] will still love them. So [the youth] also worries about his/her/their mother, their mother also worries about her child but they rarely come together to communicate, so both of them are miserable. Those who overcome that obstacle, the vast majority do very well. (Counselor)

Although many YLHIV try to hide their HIV status for the benefit of themselves and their family, this choice may actually be counterproductive.

**Impact of disclosure.** The three participants whose family knew their status mentioned family members becoming more concerned and involved with their health after disclosing:

I feel relieved to have shared. Whenever my mom calls me, she reminds me to eat nutritious foods, and asks whether I have taken the medication yet. I feel warm and loved. My sister also understands me and supports me. (MSM, 21 years old)

The PWID whose HIV status was inadvertently disclosed also received support from his family:

They will help when I'm tired. Mostly, my family members know that I was sick, so they help me more. . . Those who know it encourage me even more. No shunning, they just encourage me to try to take medicine. Because medicine helps [PLHIV] live for decades. (PWID, 24 years old)

The young man whose mother accidently discovered his HIV status, shared similarly positive experiences:

I felt that I did not have to hide anymore. . .My family has only one mother and one child. My father is not with my mother, so in the past I was very independent. My mother rarely paid attention to . . .my eating and drinking habits. In the past, she did not care. . .Since my mother learned my health condition, she has loved and cared about me more than before. (MSM, 24 years old)

This young man also served as a mental support to his mother as she accepted his HIV status:

It took my mother a week to accept my HIV status. In general, I also had to comfort my mom a lot. I studied medicine and HIV, so I know all the information very well. I provided information to my mother. . .My mother assumed that HIV could not be cured, and patients were just waiting to die, but that was not the case. Now patients can extend their

life. If we receive good medical care, we can live like a normal person. If we are relaxed mentally, everything is okay. (MSM. 24 years old)

Thus, as shown through these examples, disclosure can have a transformative, positive effect on family relationships. In preparing for the disclosure process, the counselor observed that to deal with the reactions from family, YLHIV must themselves accept their HIV status, have accurate information about HIV, and be strong mentally for others to rely on after disclosure:

Young people, from my experience. . . usually only take 3–6 months, some will take about 1–2 years, after that they are certainly very good. That means they are well aware of their situation, and ready to be strong for others to rely on. They must make sure that their parents can see that their health is good, something like, "you see, I have been infected 2 years already and still okay, even better than the old days." Maybe before, they are the person who often went to bars, played video games, and watched movies all night. But now, when they started taking the medication, they went to bed on time. They told their parents "You see, since then I haven't been going out all night and always come home on time, right?" The common idea is that once they feel confident, they are ready to share their situation. However, how quickly that process depends on each person and each individual. It is clear that after their relatives know about their situation, everything is better. (Counselor)

## Coping strategies with partners and their impact

Disclosure within romantic relationships poses specific challenges for YLHIV. Except for one participant who characterized himself as asexual, less than half (4/9) participants revealed involvement with a romantic partner. (5/9) indicated an active avoidance of romantic relationships.

**Impact of disclosure.**    Among four youth whose partners knew their HIV status, three reported positive support for their treatment and mental health from their partner.

I was sad in general, thought that not using the medicine because it was the same anyway, I already had [HIV], taking medicine didn't work. Thus, I did not take medicine. However, after my mother encouraged me and reminded me that my family had only me [as descendent], I took the medicine. At that time, my wife and my mother encouraged me to take medicine. I intended to quit using the medicine (PWID, 24 years old)

[My partner] doesn't discriminate, I am really thankful for him. He was the one who took me for testing, and to learn about treatment programs. He has been with me from the beginning till now. . . At the beginning, I was not familiar with taking medicine on time, I had to set an alarm. Despite the alarm, I forgot some days. And he is the person who reminds me regularly. . . he has accompanied me through the tests, being sick and collecting medicine, and he is still with me. (MSM, 22 years old)

I was shocked and having thoughts of suicide. But there was a guy beside me. He advised me and directed me to this center, then it passed. (Another MSM, 22 years old)

Thus, sharing status with a partner helped to relieve the burden of facing HIV alone and facilitated emotional and treatment support.

Sharing HIV status does not necessarily solve all relationship challenges, however. As one participant described,

I fell in love with a man, and I disclosed my health status. He said there was no problem, but I was feeling guilty about myself, so gradually I broke up. (MSM, 24 years old)

Thus, although his partner accepted it, the participant's guilt about his HIV status still affected their relationship.

One MSM currently not in a relationship described his strategy regarding disclosing his HIV status in personal relationships in the future:

I hide my status. When I feel the relationship is strong, promising, I will ask, similar to how I ask my friends [questions] to check their opinion about HIV. (MSM, 21 years old)

**Impact of avoidant strategies.** The fear of transmission and disclosure of HIV status to sex partners made many youth avoid romantic relationships:

Since I knew I have HIV, I have not really cared about any relationship. . . Because I am sick, I don't want to infect anyone. . .It's good for both of us. I just wait until the medicine technology is developed. (MSM, 21 years old)

I don't give myself the chance to see and have sex with anyone. For example, I spend time studying more, listening to music, cooking or participating in social activities. Whenever I have free time, I will do useful works that are good for my health and my future. I don't let the thought of seeing someone have the chance to creep into my mind. (MSM, 23 years old)

Many people text me perseveringly but I did not reply. I also narrowed myself. Because I think people probably wouldn't understand people like me. They will not like or will discriminate against such people. Thus, I usually don't get in touch with others too much. (MSM, 23 years old)

One MSM chose not disclose his HIV status with all sexual partners, but instead only carefully negotiated about condom use.

I just negotiated, unless there were some relationships that I found really close. For close relationships, I disclosed my status. I don't have many close relationships like that. Only when I get acquainted and I understand people will I talk about it. (MSM, 24 years old)

Health staff working with YLHIV had similar observations regarding avoidant strategies:

There are a lot of folks who look very handsome, have good education, their backgrounds are also very good, which means they are attractive themselves, a lot of people want to be with them but they always reject people. They don't find themselves brave enough to disclose their condition to their partner. (Counselor)

Thus, social life with romantic partners is reduced significantly for young MSM due to fear of transmission.

## Coping strategies with friends and their impact

All ten participants revealed lack of full disclosure to their social groups generally; however, a few (4/10) mentioned disclosure to small circles of friends.

**Impact of disclosure.** The smaller circles seemed to make the action of disclosing status less intimidating,

No one knows except for just 3–4 close friends. . .They are close friends in university class, we have been friends since long ago. (MSM, 23 years old)

One participant described his strategy of "testing" friends before disclosing:

With close friends, I ask something like "Now, there are many people with HIV, what do you think?" If they said something like HIV is not so terrible, if they take the medication, protect to not transmit to others. If they don't fear HIV and our relationship is close enough, I will disclose. (MSM, 21 years old)

Another youth said he knew his friends well so he foresaw that they would not reject him due to his HIV status:

I know their personalities and their reaction. They were actually quite surprised and blamed me for my lifestyle at first. . . Since then, we have kept hanging out together as normal. (MSM, 23 years old)

Sharing with even a small circle of friends offered a means of mental support for some participants. As one related,

They also encouraged me to live and think more positively. Since then, we have kept hanging out together as normal. Other people stay away from me. . . they do not dare to share furniture or sit near me [if they know I have HIV]. But my friends are still very comfortable, we still sit close to each other normally, still eat together and share furniture. . . (MSM, 23 years old).

Friends served as supports for some YLHIV. One MSM described staying with his friend when he first took ARVs and was experiencing side effects. His friend supported him psychologically (MSM, 22 years old). Disclosure to friends helped some youth relieve the pressure from hiding:

Actually, it is the same as when I revealed my sexual orientation I feel I am living more comfortably and openly, without being confined. Like, when my friends saw me taking medicine, they usually asked what kind of medicine, and I just said it was a tonic. But if my friends know my situation, they know it's ARV, I feel more comfortable. I don't need to avoid anything. . .[or] hide what medicine I'm taking. (MSM, 23 years old)

**Impact of non-disclosure.** Similar to dynamics with families, fear of stigma and discrimination hindered disclosure of HIV status to friends, even close friends:

Some of my friends discriminate against people with HIV, so I don't tell anyone. (MSM, 21 years old)

Non-disclosure also made some youth suffer from the pressure of hiding, as one described:

I feel unstable, insecure, uncomfortable when I have to hide from others for example when people ask what medication I am taking, (MSM, 21 years old)

## Coping strategies with peer groups and their impact

Three of ten participants reported that they had disclosed to peers in support groups. The other seven mentioned they did not make any effort to reach out to support groups at all.

**Impact of disclosure.** Those who had disclosed within peer groups described receiving mental health and treatment support to different extents:

> I know some counselors and people in similar situations, but I don't hang out with them. When I need their consultation, I send messages, make a phone call to ask for information. (MSM, 21 years old)

> I have a group of friends my age also living with HIV. They live happily, still work, study, live normally. If they can do that, I can do that. (MSM, 22 years old)

One participant even actively helped others who are living with HIV. He researched information about HIV not only for himself but also to help others:

> I have a group of friends who are infected with HIV. We hang out together and exchange information and knowledge. . . .There are people who have been treated for a while. Some people have just started treatment. I understand how the treatment is, because I have experienced it for a while, so I know what to do . . . and give them useful psychological advice. Some of my friends were very scared, just like me in the past. It's like suddenly, all our plans were collapsed because of HIV. . .I am 23 years old, but there are some younger people I know who are only 17, 18 years old. They are very young. . . .Those friends of mine were very confused and scared. . . . I give them psychological advice to make them less nervous and scared. (MSM, 23 years old)

**Impact of avoidant strategies.** Several participants who did not participate in peer support groups described a clear choice of avoidance. For example:

> Since I knew I was sick, I have self-constrained my life. That means having the necessary relationships only, not expanding, so I do not want to learn about these groups. (MSM, 22 years old)

One MSM even left the LGBT community after learning his HIV status.

> In the past, I participated in the Vietpride program, but since contracting HIV, I have left all groups and am not involved in such activities anymore. . . now my friends are all "normal". I do not keep in touch with anyone in the LGBT groups. (MSM, 22 years old)

In addition to missing out on social interactions, those youth who did not participate in support groups also did not receive potential emotional and moral support from peer groups.

## Coping strategies with workplace/school and their impact

Aside from familial, friend, and peer relationships, the threat of stigma in school and the workplace was felt enough to prevent YLHIV from sharing their status with classmates and coworkers. None reported disclosing in school or the workplace:

> I think HIV status should not be disclosed because we can maintain relationships and career. If one discloses, it's difficult to find a job, and the relationship could be broken. (MSM, 21 years old)

> . . .if I work at somewhere else which requires a blood test, they will discriminate against [me because of my] HIV. The more people discriminate, the more uncomfortable your spirit is. (IDU, 24 years old)

**Impact of non-disclosure strategies.** Participants described how disclosure within the workplace can affect an individual's HIV treatment. The participant who identified as PWID, for example, worked in a gas station, usually the night shift. For convenience, he kept the medicine in the trunk of his motorcycle where the temperature is not regulated and either told lies to colleagues about medication or took the medication in secret without drinking water. Thus, he took ARVs not as indicated which could be problematic for his health.

Hiding HIV status at work was also stressful for the participants. For example, when asked how he felt when needing to hide medication, one responded:

> Tired. Myself, when hiding something, I feel very uncomfortable. I am a person with excess energy. As an old saying goes, people have to think carefully before they speak, but sometimes I can't even think. I am very afraid of the moment when I might reveal something unintentionally. (MSM, 23 years old)

## Coping strategies in health care settings and their impact

Only three of the ten participants expressed a willingness to disclose within the healthcare setting (apart from the HIV outpatient clinic), and only for procedures that involved blood tests such as surgeries:

> Yes, [I disclose] when blood is involved. If a blood test is not required, I do not disclose that I am HIV-positive. (MSM, 23 years old)

Most of the participants either avoided or were hesitant to access other health care facilities rather than the OPC where they receive their HIV care. The reason for this was usually perceived stigma, as evidenced in the words of one youth:

> . . . At hospitals, sometimes people with HIV get ignored, but here [OPC] they are not. In hospitals, for example, sometimes the doctors have discriminatory attitudes towards people living with HIV. (Asexual, 23 years old)

**Impact of disclosure.** For the youth who disclosed their HIV status in the health care setting, their procedures were rescheduled to the end of the day, and most of health staff behaved as usual, with only some minor discrimination from some health staff.

> I had a colonoscopy once. Generally, when I went there, I told the doctor that I was infected with HIV and was having treatment. Then, they didn't have any problem either. I just had a very hesitant mentality. And normally, my health checks are pushed down to the end of the day so that other patients are not infected. (MSM, 24 years old)

Even if they are willing to disclose, youth described still feeling hesitant and embarassed to disclose their HIV status in health care settings.

**Impact of avoidant strategy.**   Many YLHIV reported avoiding accessing other health care facilities as much as possible. They chose strategies to avoid treatment, self-treat, or select health facilities far from home:

> Well, when I went to the hospital near my home and was am told to have a blood test, I was also afraid. Unfortunate was when I'm sick and have to go to the hospital. Sometimes I go to the hospital and if I'm told to take a blood test, I leave.. (IDU, 24 years old)

One participant shared that he made double payments for health insurance as a way to avoid disclosing his HIV status within the health system.

> I am hesitant to visit other health care facilities. I bought two health insurances. For HIV care, I use the health insurance I bought from the OPC, and I avoid going to other public hospitals. When I have other health problems, I go to private clinics. (MSM, 22 years old)

The OPC physician told about a patient who had his wisdom teeth removed without informing the dentist of his HIV status.

> One of our patients went to a private clinic to have his wisdom teeth removed, but he never said he had HIV. Usually, when people go to the dentist and say that they have HIV, probably nowhere will accept these patients. It is a similar problem in medical facilities. That's why patients often hide their status. He said that his tooth extraction that day was very difficult, and he bled a lot. His blood even splashed at the doctor. After that, he said that he was HIV+. The doctor was very upset and angry. She called us to tell us that we must advise our patients to inform others of their HIV status. But the reality is that HIV status is a person's secret... We always advise our patients to inform their doctors about their status before surgeries, but some hospitals are accepting and some hospitals are not. That's why people with HIV are worried and often hide their status. (Doctor)

Thus inconsistencies among health care facilities' acceptance and reactions could be a reason for PLHIV's reluctance to disclose.

## Discussion

This qualitative study is one of the first to focus specifically on the stigma-related experiences of youth living with HIV in vulnerable groups in Vietnam. We identified several major themes related to the relationships among stigma, coping strategies and health outcomes.

Similar to prior work [17, 26, 27], internalized stigma and perceived stigma were common among the youth in our sample due to fears of causing family distress and worries about the future including health, career, marriage, and transmission to children. Enacted stigma was less prevalent, as the high degree of perceived stigma resulted in non-disclosure or very limited disclosure to trusted persons, effectively minimizing opportunity for enacted stigma to occur. The fear of rejection and discrimination, common across studies in reviews [26, 27], was significant in our sample of primarily MSM. For MSM, HIV stigma does not exist in isolation, but coincides with the widespread stigma towards this population group in Vietnamese culture. The dual stigmatization of homosexuality and HIV and the common practice of non-disclosure have been observed in other studies with MSM living with HIV in Vietnam [34, 46]. MSM may feel pressure to have multiple sexual partners and be aware of the greater risk of

HIV transmission through anal sex. Disclosing their status could reduce the possibility for meeting their sexual needs as well as increase discrimination towards them from the MSM community.

Quite different from the experiences found in previous studies [26, 27, 50],Vietnamese YLHIV face other cultural pressures arising from collectivism and Confucianism, as they perceive responsibility to uphold the social norms to "maintain face for the family" and to have descendants, especially if they are male. These cultural characteristics may contribute to worsening of their internalized and perceived stigma given the difficulties of meeting these cultural norms in light of their sexual orientation as well as their serostatus.

To cope with internalized and perceived stigma, many YLHIV opted for non-disclosure strategies, involving hiding medication and dosetaking and keeping things to themselves, condom use negotiation in casual relationships, consistent with studies of YLHIV in other countries [17, 26, 27]. Thus, they gave up opportunities for receiving social support for the sake of avoiding negative reactions from others. In turn, low social support could affect adversely their mental health and ARV adherence and engagement in care. These findings are consistent with research which has found that non-disclosure is stressful and can worsen mental healt [25] which is already negatively affected by internalized and perceived stigma [2].

In contrast, youth in our sample who disclosed their HIV status either voluntarily or intentionally to families, friends, partners, and/or peers received good emotional and treatment support. Several said that they felt relieved from hiding. Yet as noted by health staff, in order to disclose to family, YLHIV need time to prepare mentally and be emotionally strong so that people can rely on them, gain good knowledge of HIV, and prove that they are living well despite HIV. The YLHIV who supported his mother as she learned to cope with his HIV status demonstrated exactly the same approach. Coping self-efficacy has been associated with fewer depressive symptoms and better ART adherence among PLHIV [51, 52]. Moreover, coping self-efficacy may buffer the relationship between stigma and health conditions among MSM living with HIV [53]. A feasibility study of an intervention involving mental health and coping self-efficacy approaches showed a significant improvement in psychological wellbeing [54].

The closeness of relationships with family also affected decisions about disclosure among youth in our sample, similar to findings in other studies [26]. To disclose HIV status to close relationships besides friends, testing the attitudes of close friends via hypothetical scenarios about HIV prior to disclosing could be a strategy for dealing with the fear of stigma and discrimination. A study in the US also found that YLHIV used the same strategy [55]. These findings highlight important factors related to the decisions of youth about whether to disclose.

With romantic relationships and health care (apart from HIV care), avoidant coping strategies were more common. Similar dynamics have been observed in studies with adolescents and young adults living with HIV in the United States and in the United Kingdom [24, 56]. As with non-disclosure, avoidance of close relationships may reduce social support, and further impede mental wellbeing caused by social isolation.

We found that apart from HIV care, the avoidance of accessing health care is common because of the fear of inadvertent, unauthorized disclosure of HIV status from the health care system and negative reactions from health care providers, fears common in other studies of YLHIV [26]. This finding is consistent with the findings from a qualitative synthesis about stigma, HIV and health [57]. The avoidance of accessing health care could worsen health outcomes of youth. Moreover, non-disclosure was evident in the context of blood-related procedures at dentists, and increased the risk of HIV transmission to dental staff, indicating that non-disclosure in health care settings is not only a threat to the health of YLHIV but to public health.

As a framework for developing further research in this area, we developed a model based on our findings and the published evidence base [19–24, 26, 27] which describes the

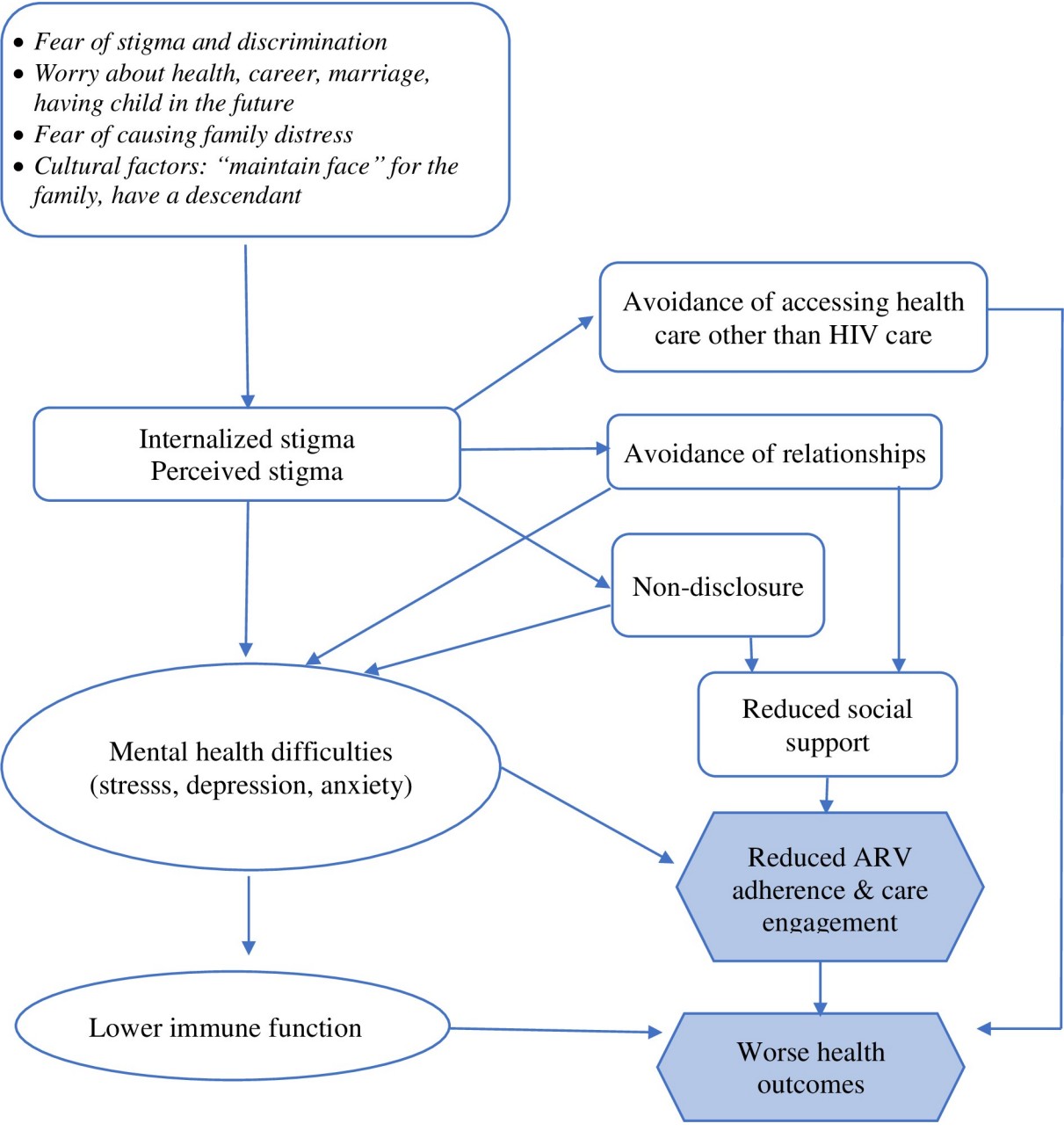

**Fig 1. Summary model for the impact of stigma for youth living with HIV in Vietnam.**

interactions between HIV-related stigma, coping strategies, and ART adherence, care engagement and health outcomes (Fig 1). The model portrays the mechanisms among HIV-related stigma, its coping strategies, and ART adherence, care engagement and health outcomes. The factors contributing to internalized and perceived stigma of YLHIV are the fear of stigma and discrimination, worries about health, career, marriage, having children in the future, the fear of causing family distress, the pressures to keep "the name" for the family, and have a descendant. To cope with these stigmas, many YLHIV used avoidant coping strategies including non-disclosure, avoidance of romantic relationships, and avoidance of health care. The

avoidant approach may have short-term benefits, but very likely, this approach impedes YLHIV's treatment and health outcomes not only short term but also long term. Social support and mental health difficulties seem to be important mediating factors in the relationships among these coping strategies and ARV adherence and care engagement. Research has found that mental health difficulties weaken the immune system [58], which in turn may negatively affect health outcomes among YLHIV. Moreover, the avoidance of accessing health care could directly affect the health outcomes of YLHIV. This model is based on the results from this sample of YLHIV in Vietnam and empirical evidence from studies among PLHIV and YLHIV, and mental health research; it will require further research for validation.

## Limitations

The study carries several limitations. First, this is a qualitative study so the findings only reflect the experiences of the participants as reported at a single point in time. Second, since nearly all of the participants were MSM, the results mainly reflect the situation of young MSM Because new HIV infections in Vietnam are disproportionately among MSM, the findings of this study are nonetheless relevant. Third, participants may have provided misleading information, due to poor recall or to social desirability bias, a common limitation of qualitative research.

## Conclusion

These interviews with youth and providers revealed important themes related to stigma, disclosure, and mental health that may inform interventions to improve outcomes for YLHIV in Vietnam and other settings. Non-disclosure with family, friends, workplaces/school, and avoidance of romantic relationships and accessing health care (apart from HIV care) are common coping strategies. Mental health difficulties and low social support seem to be the main mediating factors between these coping strategies and treatment/health outcomes among YLHIV.

Clearly, further qualitative and quantitative research with larger samples is required to validate this model for mechanisms of the impact of stigma for YLHIV. Nonetheless, a few recommendations are warranted. First, public campaigns to increase the awareness of the public related to HIV should be augmented in Vietnam in order to reduce perceived and internalized stigma and reduce risk of transmission within the community should be strengthened, Critical support for youth and their mental health should involve approaches tailored to the individual, taking into account their own context and personal capacity, including adequate time and support for psychological preparation for safe and effective disclosure. Some possible strategies for psychological preparation are suggested including: acquiring good knowledge of HIV, enhancing relationships with others, testing attitudes about HIV via hypothetical scenarios before disclosure, proving that youth are living well despite HIV, and developing coping self-efficacy skills. The development of future stigma interventions could include these strategies and examine the feasibility, acceptability and effectiveness of these strategies.

## Acknowledgments

The authors thank the youth and clinic staff in Hanoi who participated in this study for their willingness to share their personal experiences.

## Author Contributions

**Conceptualization:** Nhu Kieu Tran, Mary DeSilva.

**Data curation:** Nhu Kieu Tran, Bach Ngoc Vu, Jordan Susa, Mary DeSilva.

**Funding acquisition:** Nhu Kieu Tran, Mary DeSilva.

**Investigation:** Nhu Kieu Tran, Bach Ngoc Vu, Jordan Susa, Mary DeSilva.

**Methodology:** Nhu Kieu Tran, Mary DeSilva.

**Project administration:** Bach Ngoc Vu.

**Supervision:** Nhu Kieu Tran.

**Writing – original draft:** Nhu Kieu Tran, Bach Ngoc Vu, Jordan Susa, Mary DeSilva.

**Writing – review & editing:** Nhu Kieu Tran, Mary DeSilva.

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
