## [Decision Letter · Decision Letter 0]

4 Apr 2022

PGPH-D-21-01148

Stigma, coping strategies, and their impact on treatment and health outcomes among young men living with HIV in Vietnam: A qualitative study

Dear Dr. Tran,

Thank you for submitting your manuscript to PLOS Global Public Health. After careful consideration, we feel that it has merit but does not fully meet PLOS Global Public Health’s publication criteria as it currently stands. Therefore, we invite you to submit a revised version of the manuscript that addresses the points raised during the review process.

We look forward to receiving your revised manuscript.

Kind regards,

Rachel Hall-Clifford

Academic Editor

Journal Requirements:

1. Your co-author, Mary Bachman DeSilva - mdesilva1@une.edu ,has not confirmed authorship of the manuscript. We have resent them the authorship confirmation email; however please check that the above email address for them is correct and follow up personally to ensure they confirm. 

2. We do not publish any copyright or trademark symbols that usually accompany proprietary names, eg (R), (C), or TM  (e.g. next to drug or reagent names). Therefore please remove all instances of trademark/copyright symbols throughout the text, including TM on page 7.

Additional Editor Comments (if provided):

Thank you for this submission. We see great promise in the work, but significant revisions are necessary to meet the criteria of the journal and reviewers. Most importantly, deeper engagement with the relevant literature on HIV, sexuality, and gender in contemporary Vietnamese society is essential. Further, clarity on qualitative methods, sample, and greater representation of the depth of qualitative data gathered.

Reviewers' comments:

Reviewer's Responses to Questions

**Comments to the Author**

1. Does this manuscript meet PLOS Global Public Health’s publication criteria? Is the manuscript technically sound, and do the data support the conclusions? The manuscript must describe methodologically and ethically rigorous research with conclusions that are appropriately drawn based on the data presented.

Reviewer #1: Yes

Reviewer #2: Partly

2. Has the statistical analysis been performed appropriately and rigorously?

Reviewer #1: N/A

Reviewer #2: N/A

3. Have the authors made all data underlying the findings in their manuscript fully available (please refer to the Data Availability Statement at the start of the manuscript PDF file)?

Reviewer #1: No

Reviewer #2: Yes

4. Is the manuscript presented in an intelligible fashion and written in standard English?

Reviewer #1: Yes

Reviewer #2: Yes

5. Review Comments to the Author

Reviewer #1: This article presents qualitative data on the experience of stigma among members of an understudied and undercounted, but nonetheless quite important population: young men living with HIV in Vietnam. Its findings offer a useful contribution to future public health interventions. For the most part, it is well prepared, logically laid out, and clearly written. Its methods are very clearly described. And despite the small N, which the authors acknowledge, the work presents important findings.

Suggestions for revision:

- Probably my most major suggestion: I would like to suggest that the authors consider revising this manuscript by developing a deeper engagement with the relevant literature on HIV, sexuality, and gender in contemporary Vietnamese society. There is very little by way of a literature review in this essay. At the same time, there is quite an extensive research base on these topics that has not been consulted fully; as a result, the study findings feel somewhat thinly contextualized. I would recommend that the authors read published works by Le Minh Giang, Alfred Montoya, Pauline Oosterhoff, and others, and actively include these in the literature review as points of reference.

- Secondarily, I might also note that many of the quotes are very short and do not give a strong sense of the respondents' circumstances or worldview. Given that this is a qualitative study, it would be meaningful to have more expanded interview quotations. Perhaps it would be possible to expand some of these quotes.

- It would be useful to have a stronger sense re: whether these are entirely novel findings, or whether similar trends have been observed in other parts of the world. For example, on p. 19 the authors write that "social life with romantic partners is reduced significantly due to fear of transmission." Is this found in other global settings?

More minor suggestions:

- I might note that it would be useful to establish the ethnic identity and/or the socioeconomic status of the youth respondents, as this is not made clear and may supply useful context.

- It seems that there is a bit of a discrepancy between the statement that "the experience of internalized stigma appeared somewhat dependent on (...) personal preparedness" (9) and the interview quote that follows it on p. 10. That quote is interpreted as suggesting that it is not so much personal preparedness, but the youth's sense of not being responsible, that limited his experience of stigma. Perhaps this discrepancy could be resolved so the findings are better described.

- I would suggest that the authors' finding regarding the possible need for better infection control protocols in health care --given the possibility that some HIV+ individuals, like the one quoted interview respondent, may not disclose HIV status to care providers -- is a bit of a tangential finding. While of potential importance, it should be subordinated to the findings that *directly* address the topic of stigma. Listing it as finding #2 of 3 findings somewhat distracts from the focus of the research.

Finally, there are some line-level instances of unclear language or typographical errors, etc.:

"their highly perceived stigma resulted in not fully ..." (11) - rephrase for clarity

"PHUC-blessed" (12) - Vietnamese language does not need to be in all caps (and usually is set in italics and [square brackets]

"good luck or 'PHUC'" (12) - same as above

"oreover" (12) - Should be "Moreover"

"a social evil" (12) - This should be explained and further discussed in a footnote or a parenthetical comment, as many readers will not be familiar with the concept of "te nan xa hoi" in Vietnam

"MSM who may want multiple sexual partners" (13) - maybe this could be better described (it's unclear whether the MSM in question may want multiple sexual partners simultaneously or sequentially)

On p. 17, there is a paragraph discussing "less than half (4/9) participants revealed involvement with a romantic partner" and "of the seven single participants, most (4/7) indicated an active avoidance of personal relationships" -- the numbers are not adding up fully clearly here, and too, "personal relationships" is not clear. Does this mean "sexual or romantic relationships"?

On p. 20, the respondent is quoted saying "I revealed my gender," but I believe this should be translated as "I revealed my sexual orientation."

On p. 22: "interations" instead of interactions

On p. 23: "motorcyle" instead of motorcycle

On p. 26: "The fear of rejection and discrimination ... was severe" - How is "severe" being defined?

On p. 26: "descendents" instead of descendants

On p. 26: "in previous studies" - they should be cited here

On p. 26: "in light of their gender" instead of "in light of their sexual orientation"

On p. 26: "maintain face for the family and have descendents" - the second quotation mark (") is missing

On p. 27: "adversedly" instead of adversely

On p. 27: Reference #14 does not appear appropriately scholarly and should be replaced

On p. 27: "advertenly" - should be intentionally

On p. 27: "The YLHIV who supported his mother to cope with his HIV status did showed the same approach" - should be used the same approach

On p. 27: "todisclose" - should be to disclose

On p. 28: "the published evidence base" - which evidence specifically is being cited?

On p. 28: "a descendent" - should be a descendant

On p. 29: Limitations should include a comment about the study being cross-sectional, only drawing data from a single point in time

On p. 29: "the results likely reflect the situation of young MSM more than other groups" - it would make sense to rephrase this, because the results *definitely* reflect the situation of young MSM and not that of other groups

References - As stated, I am not sure that reference #14 appears adequately scholarly. Again, the references appear to be on the limited side and to feature very little from the extensive public health and social science scholarship on Vietnam. These works and their findings should be incorporated and should inform the arguments presented in this essay.

Reviewer #2: Comments to the Authors

This manuscript explores Stigma, coping strategies, and their impact on treatment and health outcomes among young men living with HIV in Vietnam. These data are important. Below are some comments which will hopefully help the authors in strengthening their manuscript.

Introduction

Overall:

There is a lack synthesis of literature globally, in context of Asia and Vietnam about HIV-stigma towards YLHIV. Similarly, there is a lack synthesis of the existing literature on HIV-related coping strategies that have been used by YLHIV.

Authors need to provide these in order to support their claim about the gap in knowledge they are trying to bridge with this study.

Authors really need to provide clear definition of these terms: enacted stigma (it is about both stigmatising and discriminatory attitudes and behaviours/acts); perceived stigma; and internalised/self-stigma.

I suggest these current publications to be looked at:

1. HIV Stigma and Moral Judgement: Qualitative Exploration of the Experiences of HIV Stigma and Discrimination among Married Men Living with HIV in Yogyakarta

2. Stigma and Discrimination towards People Living with HIV in the Context of Families, Communities, and Healthcare Settings: A Qualitative Study in Indonesia

Also, be consistent in using these terms throughout the manuscript.

“The prevalence of adverse mental health outcomes among YLHIV is higher than their counterparts not living with HIV”

• What is the percentage or number? Provide it.

“The fact that new HIV infections are rising among men in young age groups is

troubling, and may foreshadow further problems related to stigma”

• What is the age range?

• What is the percentage/ number new HIV infection that is rising?

The extent and experience of stigma among younger men living with HIV is not well

documented, however, especially in limited resource settings and in Asian cultures, where HIV-related stigma is particularly pervasive in part due to societal emphasis on the collective and cultural values (9).

• Authors need to provide more references about stigma in other Asian countries to support their claim that stigma is pervasive within communities in general in Asia

• You can use two sources above, and in addition, this one: “HIV stigma and discrimination: perspectives and personal experiences of healthcare providers in Yogyakarta and Belu, Indonesia”

Data Collection

Who were the interviewers? Were they some of the authors? Authors need to be transparent about this.

What was the range duration of the interviews?

How did you decide the number of participants in your study? Whether data saturation was reached? How did you determine data saturation?

I suggest the authors to use COREQ checklist to guide the report of the methods section.

Analytic Method

Who did the transcription, translation and coding or data analysis? Were they the interviewers?

Authors need to provide information about the role of the authors in this research.

It is indicated that interviews were conducted by multiple interviewers. How do you solve any discrepancies, if any, in your interpretation about the data during data analysis?

What qualitative data analysis framework the authors used to guide a step-by-step of data analysis which led to the final results (themes or categories) presented in the finding section?

Was the analysis deductive or inductive or both?

Findings

Overview

What is the rationale behind choice of only male youth in this study?

General Experience of Stigma

Internalised Stigma

The first two paragraphs under "Internalised Stigma" section are not about internalised stigma but psychological/mental health impacts of HIV. These need to be revised.

Internalised/self-stigma refers to the degree to which PLHIV endorse negative perceptions, beliefs, feelings associated with HIV and apply these to themselves.

I suggest the authors to provide clear definitions about internalised/self-stigma, perceived stigma and enacted stigma in the introduction and use them to guide the presentation of their findings. Your narratives/synthesis and the quotes that support your synthesis should reflect these concepts.

Enacted stigma:

No need to report as it was not experienced by your participants.

Dynamics underlying internalised stigma and perceived stigma:

Again, this section is mainly about HIV-related psychological challenges/impacts participants experienced.

Authors need to carefully look at this and revise, probably combine it with the first two paragraphs in the internalised stigma section under a new section about psychological impacts of HIV.

Discussion

Authors discussed the findings very well

Limitations

Generalisability is not an aspect to be achieved in qualitative research (it is typically for quantitative research), so it is not a limitation of qualitative studies.

6. PLOS authors have the option to publish the peer review history of their article (what does this mean?). If published, this will include your full peer review and any attached files.

**Do you want your identity to be public for this peer review?** For information about this choice, including consent withdrawal, please see our Privacy Policy.

Reviewer #1: No

Reviewer #2: **Yes: **Nelsensius Klau Fauk

---

## [Decision Letter · Decision Letter 1]

2 Jul 2022

PGPH-D-21-01148R1

Stigma, coping strategies, and their impact on treatment and health outcomes among young men living with HIV in Vietnam: A qualitative study

Dear Dr. Tran,

Thank you for submitting your manuscript to PLOS Global Public Health. After careful consideration, we feel that it has merit but does not fully meet PLOS Global Public Health’s publication criteria as it currently stands. Therefore, we invite you to submit a revised version of the manuscript that addresses the points raised during the review process.

We look forward to receiving your revised manuscript.

Kind regards,

Rachel Hall-Clifford

Academic Editor

Journal Requirements:

Additional Editor Comments (if provided):

Thank you for your revised manuscript. The reviewers and I agree that it is much improved be requires further attention in some areas, indicated in the attached reviews.

Reviewers' comments:

Reviewer's Responses to Questions

**Comments to the Author**

1. If the authors have adequately addressed your comments raised in a previous round of review and you feel that this manuscript is now acceptable for publication, you may indicate that here to bypass the “Comments to the Author” section, enter your conflict of interest statement in the “Confidential to Editor” section, and submit your "Accept" recommendation.

Reviewer #1: All comments have been addressed

Reviewer #2: All comments have been addressed

2. Does this manuscript meet PLOS Global Public Health’s publication criteria? Is the manuscript technically sound, and do the data support the conclusions? The manuscript must describe methodologically and ethically rigorous research with conclusions that are appropriately drawn based on the data presented.

Reviewer #1: Yes

Reviewer #2: Yes

3. Has the statistical analysis been performed appropriately and rigorously?

Reviewer #1: N/A

Reviewer #2: N/A

4. Have the authors made all data underlying the findings in their manuscript fully available (please refer to the Data Availability Statement at the start of the manuscript PDF file)?

Reviewer #1: No

Reviewer #2: Yes

5. Is the manuscript presented in an intelligible fashion and written in standard English?

Reviewer #1: Yes

Reviewer #2: Yes

6. Review Comments to the Author

Reviewer #1: (No Response)

Reviewer #2: Comments to authors

Overall, I am happy with the revision provided by the authors. Just a few small suggestions to add and can be published.

Analytic methods

“We used a thematic ...... in the interview guide.”

Comment:

• I agree, but as you haven’t provided any reference, please use this current literature as a reference to support this, look at the data analysis section that applied same kind of analysis:

Cultural and religious determinants of HIV transmission: A qualitative study with people living with HIV in Belu and Yogyakarta, Indonesia. PLoS ONE 16(11): e0257906. https://doi.org/10.1371/journal.pone.0257906

Enacted stigma

All participants revealed ...healthcare setting.

Comments:

• I don’t agree with the inclusion of this ‘enacted stigma’ in your results section, but if you do want to present it in the results section, then it is not proper to jus write the above sentence. You need to walk the reader through the story why it was not the case in your current participants and support your narratives with quotes from your participants.

I suggest, you take it our from results section and address it in the discussion section to inform the readers why it was not the case in your participants

Discussion

Quite different from the experiences found in previous studies (26,27),... sexual orientation as well as their serostatus.

Comment:

• I suggest the authors use this current literature review about stigma in Asia to support this:

Psychological and Social Impact of HIV on Women Living with HIV and Their Families in Low- and Middle-Income Asian Countries: A Systematic Search and Critical Review. Int. J. Environ. Res. Public Health 2022, 19, 6668. https://doi.org/10.3390/ijerph19116668

Limitations

“….this is a cross-sectional study…..”

Comments:

• Please be careful with the terms, “cross-sectional” is typical quantitative study design (not qualitative design). Please change it and just simply use “qualitative study”.

7. PLOS authors have the option to publish the peer review history of their article (what does this mean?). If published, this will include your full peer review and any attached files.

**Do you want your identity to be public for this peer review?** For information about this choice, including consent withdrawal, please see our Privacy Policy.

Reviewer #1: No

Reviewer #2: **Yes: **Nelsensius Klau Fauk

---

## [Editor Report · Decision Letter 2]

15 Aug 2022

Stigma, coping strategies, and their impact on treatment and health outcomes among young men living with HIV in Vietnam: A qualitative study

PGPH-D-21-01148R2

Dear Dr. Tran,

We are pleased to inform you that your manuscript 'Stigma, coping strategies, and their impact on treatment and health outcomes among young men living with HIV in Vietnam: A qualitative study' has been provisionally accepted for publication in PLOS Global Public Health.

Best regards,

Rachel Hall-Clifford

Academic Editor
